# Original Research Article

Turgor; Water Exchange; Stomatal Guard Cell;
System Dynamics Model; Cell Wall Mechanics.

**Author for correspondence:**
H. Yi,
E-mail: huy1@psu.edu

# Turgor pressure change in stomatal guard cells arises from interactions between water influx and mechanical responses of their cell walls

Hojae Yi[1] , Yintong Chen[2] and Charles T. Anderson[2]

[1]Department of Agricultural and Biological Engineering, The Pennsylvania State University, University Park, Pennsylvania, USA; [2]Department of Biology and Intercollege Graduate Degree Program in Plant Biology, The Pennsylvania State University, University Park, Pennsylvania, USA

## Abstract

The ability of plants to absorb $CO_2$ for photosynthesis and transport water from root to shoot depends on the reversible swelling of guard cells that open stomatal pores in the epidermis. Despite decades of experimental and theoretical work, the biomechanical drivers of stomatal opening and closure are still not clearly defined. We combined mechanical principles with a growing body of knowledge concerning water flux across the plant cell membrane and the biomechanical properties of plant cell walls to quantitatively test the long-standing hypothesis that increasing turgor pressure resulting from water uptake drives guard cell expansion during stomatal opening. To test the alternative hypothesis that water influx is the main motive force underlying guard cell expansion, we developed a system dynamics model accounting for water influx. This approach connects stomatal kinetics to whole plant physiology by including values for water flux arising from water status in the plant.

## 1. Introduction

Stomata are pores in the epidermis of plants that are flanked by paired guard cells, which deform to adjust the pore area. Plants regulate stomatal pore size to control $CO_2$ entry and water vapour loss. Stomatal kinetics in plants are manifestations of genetic regulation of and environmental impacts on the developmental patterning of guard and other epidermal cells (Spiegelhalder & Raissig, 2021) and function to control gas exchange for plant signalling, photosynthesis and evapotranspiration for water transport and regulation of leaf temperature. Understanding the molecular, cellular and biomechanical mechanisms that underlie stomatal kinetics is essential for elucidating form-function relationships in plants and for engineering stomata in crops for optimal yields and water use efficiency (Klejchová et al., 2020).

The canonical understanding of stomatal opening is that environmental and/or physiological stimuli trigger signal transduction in guard cells that induce proton export, plasma membrane hyperpolarisation, and the opening of voltage-gated ion channels, causing intracellular ion accumulation (Jezek et al., 2021; Lawson & Matthews, 2020; Roelfsema & Hedrich, 2005). This ion accumulation lowers the relative water potential inside the guard cell and drives water influx through osmosis (Dreyer & Uozumi, 2011). This water influx, in turn, increases intracellular turgor pressure, which in coordination with the deformation of the cell wall, drives the spatially constrained volumetric expansion of the guard cell (Carter et al., 2017) and thus opens the stomatal pore. A distinct set of signalling, transport, osmotic and biomechanical processes is thought to drive guard cell contraction and stomatal closure, which is not simply the reverse of stomatal opening (Cotelle & Leonhardt, 2016; Rui et al., 2017).

The biomechanical mechanisms of guard cell deformation have been studied to understand how plants regulate stomatal function. Earlier studies assumed that an increase in turgor is a necessary driving force for stomatal opening (Meidner & Edwards, 1975; Zeiger et al., 1987). Other studies hypothesised that stomatal opening results from guard cell bending and employed the theory of classical beam deflection to explain how plants regulate stomatal opening and

closing (Aylor et al., 1973; DeMichele & Sharpe, 1973; Sharpe & Wu, 1978; Shoemaker & Srivastava, 1973).

Despite an accumulation of studies of the biomechanics of guard cells and their cell walls over the past several decades, we have yet to gain a complete biomechanical perspective into how plants control the essential function of gas exchange by regulating stomatal kinetics. Guard cell deformations arise from interactions between changes in water content in the guard cell and the mechanical properties of the guard cell wall, which are challenging to measure simultaneously (Chen et al., 2021). Changes in turgor pressure in the cytoplasm, vacuole, or both can arise during these interactions.

Studies using finite element modelling (Carter et al., 2017; Cooke et al., 2008; Marom et al., 2017; Woolfenden et al., 2017; 2018; Yi et al., 2018) to simulate the mechanics of the guard cell wall and stomatal kinetics have typically used assumed turgor pressure values based on measurements from other experimental systems (Franks et al., 2001) rather than the modelled cells per se, and the predictions of those models are inherently confounded by this assumption as well as assumptions regarding cell wall properties in guard cells, for example, anisotropy and elasticity versus viscoelasticity. More importantly, this approach of using static pressure to model guard cell deformation neglects the effects of dynamic changes in guard cell volume and turgor pressure on stomatal biomechanics and does not address the knowledge gap of how turgor pressure in intact guard cells might dynamically change during stomatal opening, although evidence for these dynamic changes has recently been uncovered (Chen et al., 2021).

To address the above challenges and open questions, we propose a new conceptual framework for the mechanisms that drive stomatal opening. Instead of invoking turgor pressure as a driving force, water influx is proposed to be the driving force, given its earlier position in the sequence of events that underlie stomatal opening. We postulate that this approach can more effectively integrate physiological aspects of stomatal opening with biomechanical influences on stomatal kinetics. The system dynamics model described below demonstrates how this framework can effectively simulate water influx, guard cell expansion, stomatal opening, and resulting changes in turgor pressure while accommodating different potential cell wall properties.

## 2. Methods

To simulate turgor pressure arising from water influx driven by the difference in water potential, we developed a system dynamics model (Hannon & Ruth, 1997; Karnopp & Rosenberg, 1975) using Xcos (Scilab, 2021). Considering the nature of osmotic water exchange (Diekmann et al., 1993; Kramer & Myers, 2012; Oster & Peskin, 1992), water influx is modelled to follow a Maxwell–Boltzmann function, assuming that the water influx by osmosis follows diffusion.

Water influx is modelled with an increment of 1 s, then the expansion of the guard cell is determined by the instantaneous mechanical deformation of the guard cell wall following the stiffness of the wall. At this moment, the increment of guard cell expansion will reduce the turgor, which will be at least partially compensated for by additional water influx, the rate of which is modulated by the changing osmotic potential difference. The extra water influx is accounted for iteratively to complete the 1 s time step. It should be noted that this time step does not imply that water influx, volume change, and turgor change occur in series. We chose an 1 s time step as a small enough increment to allow

for simulation of simultaneous water influx, guard cell expansion, and turgor change. Such simultaneous events coincide with the observation of Baskin (2015) that oscillatory plant cell growth is not experimentally supported even when considering iterative water exchange, turgor change, and cell expansion.

Four different biomechanical behaviours of the guard cell wall were modelled to simulate the volumetric deformation of the guard cell and resulting turgor pressure. They are (a) soft guard cell wall, (b) rigid guard cell wall, (c) linear elastic guard cell wall and (d) viscoelastic guard cell wall. The soft and rigid guard cell wall hypotheses are idealistic extremes. Although these hypotheses are not realistic, extreme hypotheses reveal the implications of different biomechanical stiffnesses for guard cell walls on resulting turgor pressures.

An elastic guard cell wall hypothesises a simplistic biomechanical behaviour of the guard cell wall. It is highly likely that the guard cell wall behaves at least partially inelastically since plant cell walls in general show viscoelasticity (Baskin, 2017). However, the elastic guard cell wall will help explain the effects of cell wall constraints on turgor pressure changes during stomatal opening.

A relationship between the internal pressure of liquid and container volume is described by the definition of bulk modulus, $K$, where $V$ equals volume, $p$ equals pressure, $V_0$ equals the original container volume, and $V_n$ equals the new container volume:

$$K = -V \frac{dp}{dV}. \tag{1}$$

When stomata are stimulated to open, water flows into the guard cells, which induces an increase in guard cell volume (Franks et al., 2001; Meckel et al., 2007; Rui & Anderson, 2016). Additional water influx is required to maintain a given pressure as the guard cell expands. Assuming a simplified cylinder with internal diameter ($d$) and thickness ($t$), made of a homogeneous, isotropic, elastic material with Young's modulus ($E$) and Poisson's ratio ($\nu$), total internal guard cell volume ($V_t$) as it relates to original guard cell volume ($V_0$) with the addition of liquid with a bulk modulus ($K$) can be described using the following equation (Hearn, 1997):

$$V_t = \frac{3pd}{4tE} (5 - 4\nu) V_0 + \frac{pV_0}{K}. \tag{2}$$

Note that $E$ and $\nu$ represent the elastic properties of the guard cell wall. In the case of the rigid wall hypothesis, $E$ is set to infinity to make the guard cell wall non-deformable, which makes this equation identical to the bulk modulus definition. In the soft wall case, the guard cell volume change will be the same as the amount of water influx without any change in wall stress ($V_t = V_n$). Equation (2) applies to the elastic wall hypothesis, where $E$ is set to a specific, finite value. When viscoelastic behaviour is assumed, $E$ and $\nu$ should be modified to accommodate the time-dependency.

The viscoelastic hypothesis is one of many potential inelastic behaviours for guard cell walls. The guard cell wall consists mainly of cellulose, hemicelluloses, and pectins (Rui et al., 2018). A combination of crystalline molecular structure and circumferential arrangement is thought to allow cellulose to act as the major load-bearing component of the guard cell wall. On the other hand, hemicelluloses and pectins are thought to interact with the surface of cellulose microfibrils and also form a hydrated matrix that exhibits properties of a liquid or gel (Chanliaud et al., 2002; Whitney et al., 1999). A combination of solid cellulose microfibrils situated in a viscous wall matrix would make the cell wall exhibit mechanical behaviours of both a solid and a viscous liquid simultaneously

**Table 1.** Stomatal guard cell system dynamics model parameters.

| Parameter | Value | Citation |
|---|---|---|
| Initial guard cell volume | 3,691 $\mu m^3$ | Meckel et al. (2007) |
| Final guard cell volume | 4,600 $\mu m^3$ | Meckel et al. (2007) |
| Time for stomata to open | 30 min | Jezek et al. (2019) |
| Bulk modulus of water | 2.2 GPa | Wagner and Pruß (2002) |
| Instantaneous modulus of guard cell wall | 30 MPa | Chen et al. (2021) |
| Equilibrium modulus of guard cell wall | 20 MPa | Chen et al. (2021) |

(Baskin, 2017; Cosgrove, 2018). Viscoelasticity models describe such materials as exhibiting delayed deformation.

The Kelvin model is one widely used viscoelasticity model, where a solid and a viscous matrix are coupled to exhibit identical amounts of deformation. Because the guard cell wall and its constituents should deform as a whole, but possibly at different spatial magnitudes and temporal rates, the Kelvin model could be an appropriate starting point for modelling it.

The Kelvin model describes delayed deformation as strain ($\varepsilon$) that depends on time ($t$).

$$\varepsilon(t) = \varepsilon_0 \left(1 - \exp\left[-\frac{t}{T_{\text{retardation}}}\right]\right), \qquad (3)$$

where $\varepsilon_0$ is an ultimate strain, $T_{\text{retardation}}$ is a characteristic time constant that controls the delay in the response of a material, and $\exp\left[-\frac{t}{T_{\text{retardation}}}\right]$ represents the natural exponential function that is decaying over time proportional to a characteristic retardation time ($T_{\text{retardation}}$). In the case of the guard cell wall, $\varepsilon_0$ represents the eventual wall deformation required to achieve the observed volume increase in the guard cell when the stomatal complex fully opens. $T_{\text{retardation}}$ is thought to reflect the constitution of the guard cell wall and the duration required for the guard cell wall to undergo complete stress relaxation. In this study, $T_{\text{retardation}}$ is set to be 100 min reflecting previous observations from stomatal opening experiments (Rui & Anderson, 2016; Rui et al., 2017).

The remaining parameters of the system dynamics model for stomatal opening are listed in Table 1. The time for stomata to open represents the duration required to approach maximal stomatal conductance of Arabidopsis (Jezek et al., 2019). For the viscoelastic wall model, the instantaneous modulus of the guard cell wall represents its resistance to instantaneous deformation, and the value represents the maximum $E_1$ value of the wall in Arabidopsis guard cells immediately after the induction of stomatal opening (Chen et al., 2021). The equilibrium modulus represents the resistance of the cell wall to deformation when stress relaxation is complete, and the value represents the minimum $E_1$ value of the cell wall in Arabidopsis guard cells at the end of stomatal opening (Chen et al., 2021).

## 3. Results

### 3.1. Modelling stomatal kinetics with mass conservation: Water influx following an s-curve in osmotic potential drives stomatal opening

In this model, water influx gradually increases in the beginning, followed by a rapid increase, then finally slows down asymptotically, approaching a predetermined volume. This curve represents a typical diffusive mass transfer following Fick's law. Figure 1a depicts

the cumulative volume of water flow into the guard cell during opening. The initial guard cell volume representing the closed state and the final volume representing the open state are set to be 3,691 and 4,603 $\mu m^3$, respectively, referring to the reported observation by Meckel et al. (2007).

#### 3.1.1. Simulated soft or rigid guard cell walls do not enable appropriate volume changes.
When the guard cell wall is modelled to have no resistance to tensile deformation, guard cell volume increases to accommodate water influx. This results in an unconstrained volume change in the guard cell, and turgor pressure does not increase (Figure 1b). While this result is a hypothetical verification of the model and does not conform with observations of real stomatal complexes (Chen et al., 2021; Rui et al., 2018), it differs from the canonical notion that an increase in turgor deforms the guard cell.

On the other hand, when the guard cell wall is modelled to be a completely rigid material, there is no change in guard cell volume. While guard cell volume remains at its initial value, water influx induces a rapid initial increase in turgor pressure (Figure 1c). Due to the finite bulk modulus of water, the turgor increase is not boundless but is two orders of magnitude higher than reported values (Franks et al., 1995; 1998). While this result is another verification of the model using an opposite extreme, it also differs from the canonical description of how turgor drives the expansion of guard cells. Combined with the soft wall scenario, the rigid wall model suggests that an appropriate level of wall stiffness is necessary to achieve an appropriate level of turgor increase during stomatal opening.

#### 3.1.2. Modelling the guard cell wall as a linearly elastic material enables a realistic turgor increase.
To model the guard cell wall as an isotropic elastic material, constant moduli values are assigned for Young's modulus ($E$) and Poisson's ratio ($\nu$). These values are assumed and deduced from previous studies (Chen et al., 2021; Yi et al., 2018). When stomatal opening is simulated with water influx constrained by an elastic cell wall, guard cell volume increases following water influx (Figure 1a). Accordingly, turgor increases following the same trend (Figure 1d). Due to interactions between the expanding guard cell volume and volumetric compression imposed by elastic wall, turgor pressure increases up to a value that is closer to measured values (Chen et al., 2021). It should be noted that the magnitude of turgor is almost three orders of magnitude smaller than the rigid cell wall case. Combined with the rigid cell wall result, this result suggests that the stiffness of the guard cell wall can be inferred from the turgor pressure if the amount of water influx is known. Pressure probe measurements (Franks et al., 1995; 2001; Steudle, 1993; Zhang et al., 2011) employ this principle when there is no water exchange. In addition, this simulation predicts a rapid opening of the stomatal pore in the earlier stage of stomatal opening, as discussed by Jezek et al. (2019), without the need to introduce a complex biomechanical feedback pathway via signal exchange with neighbouring cells.

#### 3.1.3. Modelling the guard cell wall as a viscoelastic material allows for rapid initial deformation.
The system dynamics simulation of stomatal opening with viscoelastic wall results in more gradual guard cell volume increase than the case of elastic guard cell wall (Figure 1). As a result, the guard cell turgor decreases from initially higher turgor following an exponential decay trend (Figure 1e). This is because a Kelvin material behaves more like an elastic material in the beginning, followed by delayed and extended strain, which will release the accumulated strain energy in the cell wall and turgor.

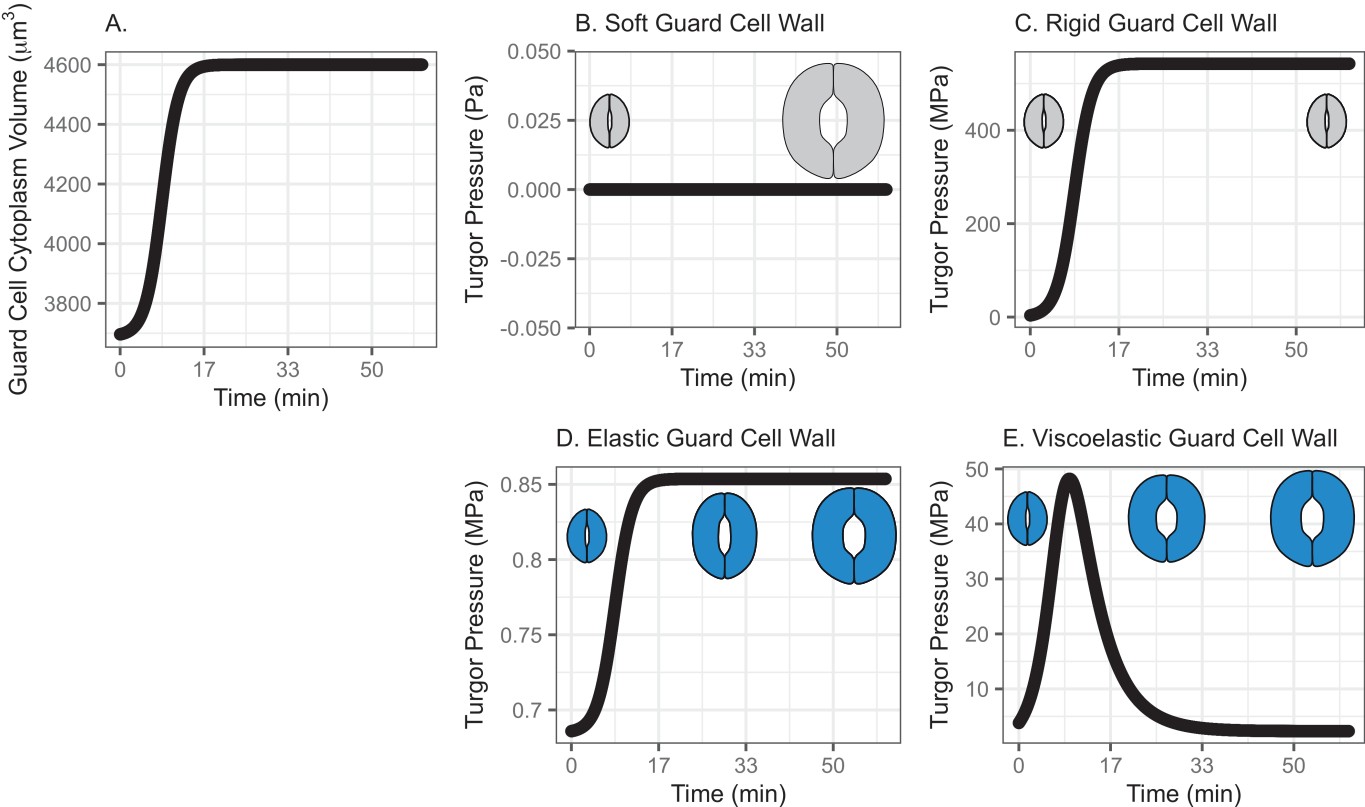

**Fig. 1.** (a) Volume change in the guard cell due to water influx following a diffusive water transfer from neighbouring cells/the apoplast Fick's law. (b) Turgor pressure from a stomatal opening simulation when the guard cell wall is assumed to have zero stiffness, that is, the guard cell wall is infinitely soft. As a result, the guard cell expands as much as the cumulative water influx and turgor does not change. (c) Turgor pressure from a stomatal opening simulation when the guard cell wall is assumed to be infinitely stiff, that is, the guard cell wall is rigid. As a result, the guard cell does not deform despite the water influx and turgor increase to a much higher value than the reported value (Franks et al., 1995; 2001). (d) Turgor change from a stomatal opening simulation when the guard cell wall is assumed to behave as an isotropic elastic material with finite stiffness, that is, the guard cell wall behaves elastically. As a result, the guard cell expands following the cumulative water influx and the turgor rises to a more reasonable level, as the dramatised illustration depicts. (e) Guard cell turgor during simulated stomatal opening when the guard cell wall is assumed to behave as a Kelvin viscoelastic material with finite stiffness, that is, the guard cell wall behaves viscoelastically with finite and constant stiffness. As a result, the guard cell expands more in the earlier stage and develops a higher turgor level, followed by a much smaller guard cell volume expansion with turgor decreasing to a value comparable to those reported (Franks et al., 1995; 2001) as the dramatised illustration depicts.

In the elastic model of the guard cell wall, the instantaneous behaviour of the elastic wall is tracked over time. At every time step, the elastic deformation of the guard cell wall is instantaneous. In the viscoelastic model of the guard cell wall, time-dependent relaxation (dampening) accumulates over time. This makes the overall guard cell behaviour quite different in elastic and viscoelastic models. This difference produces different calculations of turgor values over time.

In the viscoelastic model, the predicted turgor pressure peaks at approximately 47 MPa, which is an order of magnitude higher than the reported values (Chen et al., 2021; Franks et al., 2001). The high turgor pressure develops at the earlier stage and quickly begins to decrease to close to 1 MPa, which is in the same order of magnitude as the reported values (Chen et al., 2021; Franks et al., 2001). This gradual decrease in turgor pressure is the combined result of viscoelastic guard cell wall deformation and a decreasing rate of water influx.

### 3.2. Water influx following logarithmic growth in osmotic potential drives stomatal opening

While maintaining the assumption that water influx is driving guard cell wall expansion and stomatal opening, the pattern of water influx can follow different shapes with different values of Maxwell–Boltzmann function parameters. Figure 2a shows water influx following such a logarithmic growth pattern that rapidly increases in the beginning, then slows down asymptotically approaching a predetermined volume. The initial guard cell volume representing the closed state and the final volume representing the open state are set to be 3,691 and 4,603 $\mu m^3$, respectively, referring to the reported observations by Meckel et al. (2007), which is consistent with the previous simulation.

**3.2.1. A soft or rigid guard cell wall does not induce an appropriate turgor increase.** When the guard cell wall is modelled to have no resistance to tensile deformation (soft) or as a completely rigid material, the water influx rate shown in Figure 2a does not result in realistic changes in turgor pressure. A soft guard cell wall results in no change in turgor pressure (Figure 2b), whereas a rigid guard cell wall results in a maximum turgor pressure (Figure 2c) that is three orders of magnitude higher than reported values (Franks et al., 1995; 1998). Similar to the result with the previous pattern of water influx (Figure 1), these extreme models demonstrate that an appropriately compliant guard cell wall is necessary for an appropriate level of turgor increase during stomatal opening.

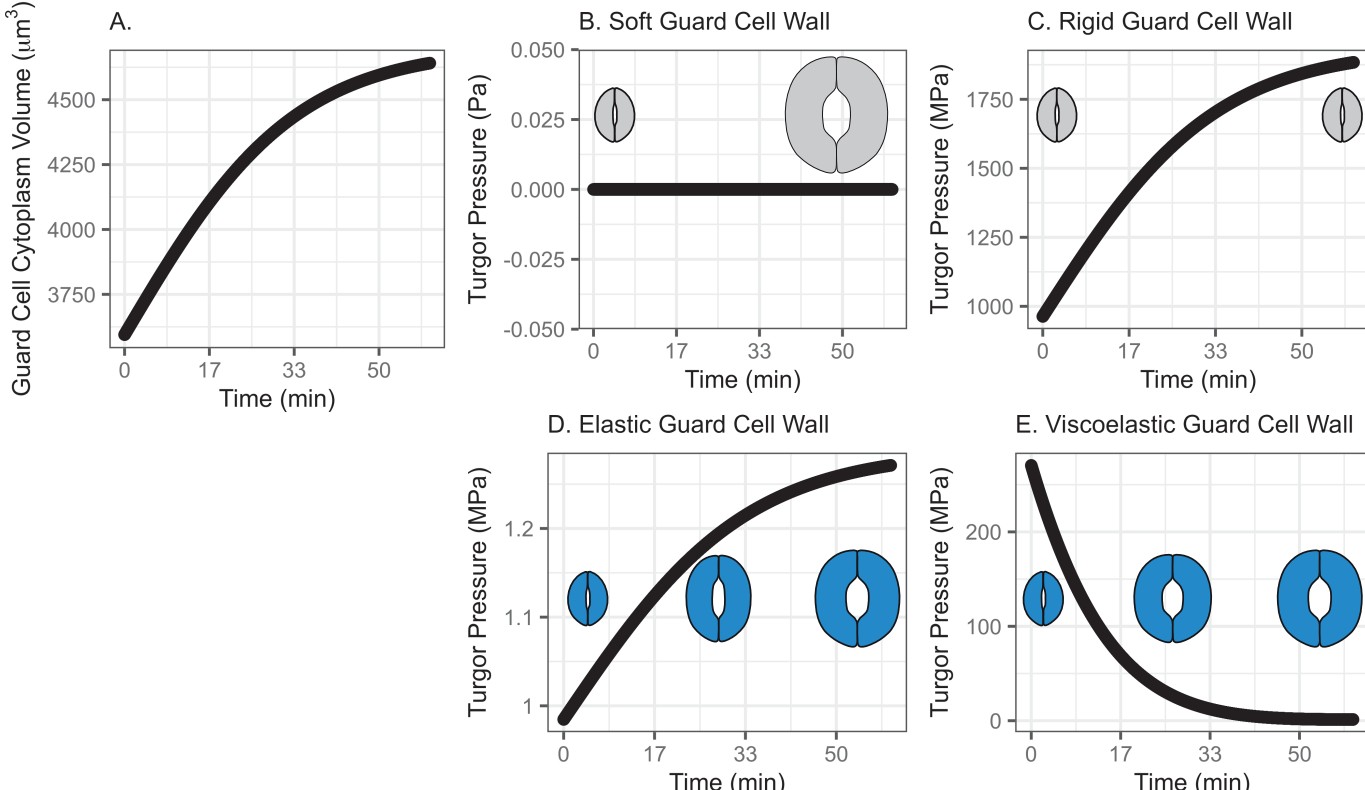

**Fig. 2.** (a) Volume change in guard cell due to water influx following a Maxwell–Boltzmann function arising from osmosis as the driving mechanism of water exchange between neighbouring cells/the apoplast and the guard cell, when the water influx is assumed to follow a pattern similar to a growth function and does not have an initial low influx rate. (b) Guard cell turgor pressure from a stomatal opening simulation when the guard cell wall is assumed to have zero stiffness. While guard cell expands as much as the cumulative water influx, turgor does not increase as the dramatised illustration depicts. (c) Guard cell turgor pressure from a stomatal opening simulation when the guard cell wall is assumed to be infinitely stiff. As a result, guard cell turgor increases gradually following the water influx pattern. (d) Guard cell turgor change from a stomatal opening simulation when the guard cell wall is assumed to behave as an isotropic elastic material with finite stiffness. When guard cell expands proportionally to the cumulative water influx, turgor increases gradually following the water influx pattern. (e) Guard cell turgor during simulated stomatal opening when the guard cell wall is assumed to behave as a Kelvin viscoelastic material with finite stiffness. Guard cell turgor starts at a much higher level followed by a rapid decrease that resembles an exponential decay approaching a turgor level similar to the reported value of 5 MPa (Franks et al., 1995; 2001).

### 3.2.2. Modelling the guard cell wall as a linearly elastic material results in a monotonic turgor increase.

Similar to the previous water-influx pattern (Figure 1), when guard cell wall expansion is simulated with water influx constrained by an elastic cell wall, guard cell volume increases following water influx, and turgor increases following the same trend (Figure 2d) up to 1.3 MPa, which is close to the reported values (Chen et al., 2021; Franks et al., 2001). In other words, guard cell turgor increases monotonically with a steady water influx throughout the stomatal opening when the guard cell wall behaves as a linear elastic material.

### 3.2.3. Modelling the guard cell wall as a viscoelastic material results in an instantaneous high turgor pressure.

Consistent with the *S*-shaped water influx model, $T_{\text{retardation}}$ is set to be 100 min reflecting previous observations of stomatal opening experiments (Rui & Anderson, 2016; Rui et al., 2017). The system dynamics simulation of stomatal opening with viscoelastic wall results in the guard cell volume increasing similarly to the *S*-shaped water influx model (Figure 1). Because of a higher rate of water influx than the *S*-curve in the initial stage, the guard cell turgor increases instantaneously, and a viscoelastic wall (Figure 2e) results in exponentially decaying turgor. Again, the decreasing trend of turgor pressure is due to the continued guard cell wall extension, which is a characteristic creep of a Kelvin material. The comparison between the turgor pressure trends of the *S*-curve and the logarithmic growth water influx

suggests that turgor change during stomatal opening is sensitive to the water influx rate at a very early stage of opening.

## 4. Discussion

### 4.1. Pressure increase is achieved either by the addition of liquid or by a decrease in volume, but guard cells do not diminish in volume during stomatal opening

Assuming that the guard cell wall expands proportionally to turgor increase, the observed linear trend of stomatal pore opening (Rui & Anderson, 2016) suggests linear increase in turgor pressure. For turgor pressure to drive stomatal opening in this way, it should increase proportionally to water influx. These relationships appear to be intuitively correct. However, a closer look at the bulk modulus definition (Wagner & Pruß, 2002 and equation (1)) does not explain the presumed relationship between turgor and guard cell expansion. According to the bulk modulus definition, the guard cell volume must decrease for turgor to increase.

This contradiction appears to arise from the assumptions of the bulk modulus definition, that is, an infinitesimal volume change and the conservation of mass. Not only does the amount of change in guard cell volume during stomatal opening becomes too large to apply this relationship, but also stomatal opening accompanies or is driven by water exchange, which violates the conservation of mass. Therefore, when guard cell expansion and turgor are to

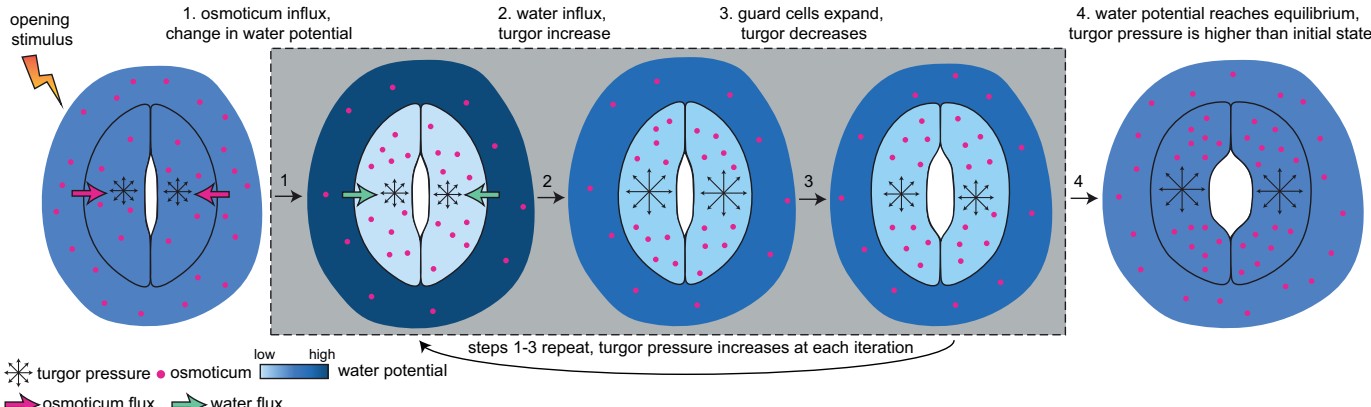

**Fig. 3.** Illustration of mechanical processes involved in stomatal opening. Stomata at resting closed state have balanced water potential with only mildly higher osmotic potential in guard cells than in the surrounding environment and thus a moderate level of turgor pressure in guard cells. Upon sensing stimuli, 1, water potential rises in guard cells and drops in surrounding environments due to the transfer of solutes from the surrounding environment to guard cells; 2, The water potential difference between guard cells and the surrounding environment leads to water influx and a transient turgor pressure increase in guard cells; 3, The resulting increase in guard cell volume leads to a small degree of stomatal opening and releases some turgor pressure inside the guard cells. This water potential difference-caused 'water influx, turgor increase, cell expansion and mild turgor drop' process (1, 2 and 3) repeats until 4, water potential reaches an equilibrium between guard cells and pavement cells, and turgor pressure eventually increases and reaches a plateau in guard cells. It should be noted that the systems dynamics model developed in this study predicts changes in turgor pressure and guard cell volume change and not the stomatal pore size.

be determined, it is essential to use appropriate governing laws that consider water flow or the extra volume of water. Because the interest here is the quantity of water influx into the stomatal guard cell, it seems to suffice to employ the conservation of mass with source and sink of water rather than focusing on the water flow.

As water flows into guard cells and they also expand, increases in guard cell volume negate the increasing turgor pressure, much like filling a water balloon does not substantially increase the pressure of the water in the balloon. Because the guard cell expands during stomatal opening (Meckel et al., 2007; Yi et al., 2018), any pressure increase can be achieved only by the addition of water. The observed turgor increases during stomatal opening (Chen et al., 2021) are a result of water influx and biological control of the mechanics of the guard cell wall.

This relationship implies more complex pressure dynamics during guard cell deformation than have been previously assumed by positing linear correlations between water influx, turgor pressure, and guard cell expansion (Rui & Anderson, 2016), which in fact are not evident when guard cells are artificially pressurised (Franks et al., 2001). Although Franks et al. (2001) used the power function to model the relationship between guard cell turgor pressure and volume, this does not explain the temporal trend of guard cell turgor or volume change during stomatal opening.

This study shows that the temporal trend of and the relationship between guard cell turgor and volume are accurately modelled and simulated with system dynamics modelling. The results suggest that the magnitude and trend of the guard cell turgor change over the course of stomatal opening result from the combined effects of the trend of continued water influx and the mechanical response of the guard cell wall. Accepting that water influx is the driving mechanism of stomatal function, we can connect stomatal kinetics, as a result of mechanical responses in the guard cell wall, with physiological regulation. For example, modelling stomatal opening as being driven by water influx provides relevant research questions, such as the physiological control of osmotic potential and guard cell wall permeability (Hachez et al., 2017; Roelfsema & Hedrich, 2005; Sussmilch et al., 2019), the quantification of which will lead to more accurate estimation of water exchange patterns during stomatal opening and closing.

We believe that the presented modelling framework allows for connecting the physiological aspects of stomatal function and the biomechanics of the guard cell wall. For example, the effect of the change in the water permeability of guard cells through the regulation of ion channels, pumps and aquaporins can be investigated quantitatively. Likewise, the development and control of the osmotic potential difference between a guard cell and pavement cells can be studied with a quantitative connection to stomatal kinetics and conductance. To that end, it is critical to quantify osmotic potential during the stomatal opening to probe how water exchange underlies this process. Then, the osmotic influx and efflux of water through the guard cell wall can be quantitatively modelled from previous studies (Atzberger & Kramer, 2007; Jezek et al., 2019; Kramer & Myers, 2012; 2013; Oster & Peskin, 1992; Peng, 2012).

Currently, a knowledge gap exists concerning how turgor arises from physiologically controlled water movements in and around stomatal complexes. A simple mathematical model of turgor pressure (as the imposed force) and guard cell deformation (as the resulting deformation) is insufficient to effectively describe stomatal kinetics given the non-trivial changes in guard cell geometry and volume that are observed using 3D microscopy before and after induced opening (Meckel et al., 2007; Yi et al., 2018). The potential for feedback between ion transport, water influx, increases in turgor pressure, alterations in wall mechanics, and volume changes in guard cells calls for investigating stomatal kinetics using a system dynamics approach (Chen et al., 2012; Efendiev et al., 2015; Hills et al., 2012; Jezek et al., 2019). This iterative process of water influx into the guard cell, the increase in volume, and resulting turgor pressure is illustrated in Figure 3.

Studies connecting the quantitative aspects of water transport in stomatal function (Chaumont & Tyerman, 2014; Hachez et al., 2017) are not as abundant as those analysing the role of ion exchanges. To engineer stomatal function based on quantitative understanding of the form and function of stomata, elucidating the mechanisms linking the transport capacity of guard cells to stomatal kinetics is essential. We propose that quantifying water exchange will help close a key knowledge gap in elucidating how plants regulate their stomata. System dynamics models of stomatal function provide a pathway to investigate and determine turgor and

guard cell wall mechanics, which will complement the current lack of experimental method to measure cell wall mechanics, turgor pressure, and ion and water flow simultaneously.

### 4.2. Viscoelastic or other non-linear properties are expected for the guard cell wall

The assumption that increasing turgor pressure drives stomatal opening can be explained phenomenologically using the following stepwise, causally linked processes:

1. Influx of water into the guard cell.
2. Momentary increase in turgor pressure.
3. Yielding and anisotropic expansion of the guard cell wall.
4. Volumetric, anisotropic expansion (elongation) of paired guard cells with a constraint on stomatal complex height.
5. Widening of the stomatal pore.

In reality, without accounting for additional water influx, a volume increase in the guard cell would release any momentary increase in turgor pressure. To validate the idea that turgor pressure is the ultimate driving force for stomatal opening, the mechanical responses of the guard cell wall should be described in conjunction with those regarding water influx. In other words, without a mass balance of water, a monotonic stress-strain relationship of the guard cell wall will not be able to account for the non-linear biomechanical response of stomatal opening to changes in turgor. This, perhaps, is the origin of the discrepancy between induced and simulated turgor changes and degrees of stomatal opening (Franks et al., 1995; 1998; 2001; Mott & Franks, 2001) and the discrepancy between predicted and measured gas exchange rates, especially at early stages of stomatal opening (Jezek et al., 2019). On the other hand, hypothesising that stomatal opening is driven by the forward osmosis controlled by the osmotic potential difference between guard cells and pavement cells, guard cell deformation will follow the water flow rate, which will be higher at the beginning of stomatal opening following Fick's law.

A linear force-displacement relationship, such as Hooke's law, assumes an infinitesimally small amount of deformation such that any altered geometry does not induce a large enough force boundary condition to be considered. Because guard cell volume and surface area both increase by as large as much as 25% during stomatal opening (Meckel et al., 2007), volume expansion should be accounted for in estimating turgor changes during stomatal opening.

To maintain the increasing trend of guard cell turgor during stomatal opening (Chen et al., 2021), extra water should be supplied to compensate for the pressure relieved by expansion in guard cell volume. Assuming an elastic guard cell wall achieves such a trend with appropriate consideration of extra water influx. However, none of the volume changes calculated for the trend of water influx simulated here (Figures 1 and 2) fully explain the initial rapid increase in gas exchange measured during stomatal opening (Jezek et al., 2019).

Because water influx into a guard cell is osmotically driven, it is intuitive to assume a gradual increase in water influx at an early stage of stomatal opening. To counteract such a hypothesised water influx, time-dependent guard cell wall mechanics are needed. A viscoelastic guard cell wall embodies such a time-dependent behaviour and results in the turgor change at a higher level of turgor at an early stage of stomatal opening, depending on the assumed water influx trend. In other words, the viscoelastic guard cell wall results in a non-monotonic turgor change. Although this predicted turgor decrease is not consistent with recent experimental data (Chen et al., 2021), a time-dependent guard cell wall model provides a potential mechanism for limiting turgor increases during stomatal opening.

Moreover, modelling the guard cell wall as a viscoelastic material suggests relevant research questions on the structure and biomechanics of the guard cell wall, including the values and ratios of wall moduli in different directions and sub-regions of the wall and whether those values change in response to protein-mediated modification of wall mechanics, such as that carried out by expansins (Zhang et al., 2011), during stomatal opening, closure, or both. This change in wall mechanics can be modelled as a time-dependency of the viscoelasticity, for example, $T_{\text{retardation}}$. Whereas the elastic components of the guard cell wall in this study are modelled to be isotropic, combining system dynamics models of stomata with finite element models encompassing anisotropic guard cell walls (Carter et al., 2017; Chen et al., 2021; Marom et al., 2017; Woolfenden et al., 2017; Yi et al., 2018), along with capturing additional quantitative data regarding the flow of water into and out of guard cells during stomatal dynamics, will advance the understanding of how plants achieve and control stomatal function. For example, determining parameters for material models of guard cell wall will lead to research questions regarding the structures and behaviours of guard cell walls, for example, weather time-dependent delays in biomechanical responses and/or cell deformation can be linked to the biomechanical interactions between the wall matrix and cellulose surface and how they are controlled by plants to facilitate stomatal opening and closure.

### Acknowledgements

We appreciate valuable discussions with Dolzodmaa Davaasuren, Sedighe Keynia, Dr. James Wang and Dr. Joseph Turner.

**Financial support.** This work was supported by the National Science Foundation under Grant MCB-2015943/2015947. The administrative operations of the research activities are partially supported by USDA-NIFA Agricultural Experiment Station Project 4671.

**Conflict of interest.** The authors declare no conflicts of interest.

**Authorship contributions.** H.Y., Y.C. and C.T.A. conceived the study, analysed the simulation results and wrote the article. H.Y. developed the system dynamics model and gathered simulation results.

**Data availability statement.** This study is based on a system dynamics model developed on SciLab. Relevant code can be accessed upon request to the corresponding author.

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
