## [Reviewer Report]

Dear Editor,

We are submitting this original research article, entitled “Turgor pressure changes in stomatal guard cells arise from interactions between water influx and mechanical responses of their cell walls,” to demonstrate how wall mechanics influence stomatal kinetics and to highlight the importance of water exchange in driving turgor changes during stomatal opening. Using a quantitative system dynamics simulation of guard cell expansion driven by water exchange during the stomatal opening, we present an approach that can connect stomatal kinetics to whole plant physiology by including values for water flux or exchange that arise from water status in the plant. We believe that our study would appeal to the readership of Quantitative Plant Biology with a novel perspective on how stomatal kinetics are regulated quantitatively and systematically. We look forward to your reply.

Sincerely,

Dr. Hojae Yi and Dr. Charles T. Anderson

The Pennsylvania State University

---

## [Reviewer Report]

*Comments to Author*: The manuscript presents a theoretical mathematical model for stomata deformation, using a bulk modulus approach and assuming the guard cells are equivalent to isotropic cylinders. The authors consider two different water influx scenarios and explore how turgor pressure would evolve with time, assuming elastic or visco-elastic cell wall properties for the guard cells. This model, although simplified, provides a useful first step towards understanding stomata hydrodynamics.

Before publication, I would recommend to address some issues:

Major issues:

1) The paper is very long and would benefit from being shortened. For example, the two paragraphs between lines 221 and 239 describe the same situation as the paragraphs between lines 162 and 181. I think they could be deleted without compromising the text.

2) lines 350-351: “In reality, the volume increase in the guard cell would release any momentary increase in turgor pressure, invalidating the idea that turgor pressure is the ultimate driving force for stomatal opening.”

3) This sentence would need to be revised. Turgor pressure is indeed the driving force for stomatal opening, since it is the cause of cell wall deformation in guard cells, either by an elastic or viscoelastic process. Maybe what the authors mean here is that water flux is needed to maintain turgor pressure? This is what the authors explain in lines 246-247: “guard cell turgor increases monotonically with a steady water influx throughout the stomatal opening when guard cell wall behaves as a linear elastic material.”

4) Line 156- 157 « In this model, water influx gradually increases in the beginning, followed by a rapid increase, then finally slows down asymptotically, approaching a predetermined volume. » Could the author explain the choice of this curve?

5) lines 192-194: “Combined with the rigid cell wall result, this result suggests that the stiffness of the guard cell wall can be inferred from the turgor pressure if the amount of water influx is known.” It would be worth noting that this is the principle used in pressure probe measurements and refer to studies using this method (e.g. Franks et al. [17] Plant Physiol, Zhang et al. [52] Plant Cell Rep)

Minor issues :

1) The captions for Figures 1 and 2 are the same. Their caption should mention the two different water influx scenarios that are considered for the simulations.

2) Lines 112 and 121 : Missing equation numbers

3) Lines126: replace “this equation” with “equation 2”

4) Line 146: replace exp[] with correct expression

5) Line 254: revise the sentence: “the guard cell turgor instantaneously high turgor pressure”

6) Line 243: “turgor, increases”

---

## [Reviewer Report]

*Comments to Author*: The manuscript submitted by Yi et. al., investigates the concomitant relationship between water uptake, turgor pressure increase and guard cell expansion during stomatal opening. The authors use three constitutive equations of known mechanical principles to model the relationship between increasing guard cell volume and turgor pressure. The mechanics of the cell wall were considered under the extremum values of “soft guard cell” which provides no resistance to external forces, a completely “rigid” cell wall that resists all deformation, a linearly elastic guard cell wall and a viscoelastic cell wall.

Although the approach and results are well-described, my main concern is the extent to which the paper makes a significant advance in our conceptual understanding (which is a premise of the paper). At points the manuscript seeks to create a straw man which the authors then proceed to pull apart, e.g, p270-71, or line 262 “guard cells do not diminish in volume during stomatal opening”. I am not sure that the discrepancy between the authors’ views and those of other researchers in the field is as large as suggested by the paper. In particular, the authors stress that “water influx is the driving mechanism of stomatal opening”, inferring that the focus on altered turgor pressure by other researchers is somehow an alternative or exclusive view. I am not convinced by this argument. The vast majority of researchers will accept that altered turgor pressure is driven by water flux in and out of the guard cells. As the paper highlights, the question of how water flux is translated into altered pressure will depend on a combination of wall mechanical properties and the ability of the cell membranes to transfer ions and water. I think these are important points to make, but the authors’ suggestion that they are novel is not so convincing. Toning down these statements would lead to a more balanced manuscript.

Specific points

1) In the figures the authors refer to “guard cell cytoplasm” volume. Do they really mean this? Most research has focused on the role of the vacuole, with changes in cytoplasm volume not playing a major contribution to guard cell swelling/deflation. Is this purely a mix-up in terminology? If so, then they should check/correct both figures and text. I think they mean “guard cell volume” (essentially cytoplasm and vacuole).

2) In Fig 1B and 2C the authors show that if the guard cell wall provides no mechanical resistance then any increase in volume (by influx of water) does not lead to any change of pressure. Although totally valid, I think you can come to the same conclusion by logic. The results act to validate that the model is not producing crazy outputs, so perhaps these data are best viewed as a control rather than new insight.

3) Similarly, the modelled outputs in Fig 2B and 2C where the wall is rigid and inextensible indicate a rapid increase in turgor as water flows into the cells. Again, I would view these outputs as model validation rather than showing something that would not be predicted from first principles.

4) The other two simulations are interesting, although again one can argue whether the outputs shown are non-intuitive (i.e. surprising). They do indicate that the modelled turgor pressure changes in the visco-elastic system (Fig 2) do not reflect the measured values available- which is interesting- though later in the Discussion the authors nevertheless explore this point in some detail. Also, initially, a viscoelastic material behaves elastically, which I would expect to be consistent with the preceding elastic scenario, before the viscous dampening contributes. Is this the case in there model?

5) Line 55: “Guard cell deformations arise from interactions between water flux into or out of the guard cell and the mechanical properties of the guard cell wall”. Flux is a rate, not a force. There has to be a resultant force leading to deformation. This force is clearly related to water flux, but is not the same thing, This is one example where I feel the authors are dancing on a pinhead to distinguish their ideas from those already extant in the literature. In their model the changes in flux and pressure occur simultaneously, rather than sequentially, which is an advance. Nevertheless the two are intimately entwined (via wall mechanical properties, as the authors correctly indicate). By focusing on water flux the authors usefully direct attention to our lack of knowledge on what defines the dynamics of this step, but the efforts to separate the two (flux and pressure) are, I think, overdone.

6) Line 290 “This relationship implies more complex pressure dynamics during guard cell deformation than have been previously assumed by positing linear correlations between water influx, turgor pressure, and guard cell expansion (Rui and Anderson, [38]), which in fact are not evident when guard cells are artificially pressurized (Franks et al., [17]).”

I think that most researchers in the field would totally agree with this statement, and similar ones elsewhere in the manuscript. I guess the main point here is that, in the absence of much reliable quantitative data for many of the parameters require to truly model this complex system, making simple (but probably incorrect) assumptions is a good place to start. The usefulness of the models (as in this paper, which also incorporates a host of assumptions) is that if they capture at least some element of the “real” system, then they are providing some sinsight, and if they do not then that is a strong indication that the underlying assumption is incorrect (as one could argue here for the visco-elastic model). The need for more reliable, quantitative data on a whole host of parameters (from water flux dynamics to wall mechanical properties) would be accepted by most people working in this area.

7) Line 334 “stomatal function of opening and closing is achieved by the volumetric deformation of the guard cell pair, which is driven by water exchange”

Is this a novel conclusion?

8) Line 349 “In reality, the volume increase in the guard cell would release any momentary increase in turgor pressure, invalidating the idea that turgor pressure is the ultimate driving force for stomatal opening. In other words, a monotonic stress-strain relationship of the guard cell wall will not be able to account for the non-linear biomechanical response of stomatal opening to the turgor change.”

Doesn’t cell volume increase requires relaxation of the wall, which implies a change of tension in the wall, which must come from turgor? That the ultimate cause is inflow of water is fine, but I get the impression again of dancing on pin-heads on the point of whether turgor pressure is the “ultimate driving force”. With the second sentence, I totally agree that monotonic stress-strain relationships of the wall are not able to account for a non-linear response to turgor change, but I’m not sure how this follows on from the first sentence.

9) Figure 3 is not immediately self-explanatory. Steps 1, 2 and 3 are not described. I suggest improving to make it easier to interpret.

Other comments

10) Line 25: stomatal kinetics are manifestations of genetic regulation-

Well, only very indirectly. Actual stomatal kinetics will be dependent on emergent mechanical properties of the system and the rate and extent will be dependent on environmental history of the plant.

11) It is common practice in modeling papers to provide a table of parameters and their citations (if available).

12) Hearn 1997 citation is missing.

13) The figures show cartoons of stomata opening/closing, and in the text statements are made about pore opening/closing. I think the authors need to be careful to indicate that their model does not directly give pore size as an output. This is already in the manuscript, so this is more a case of making clear that indications on pore size are estimations/interpretations, not modelled (unless I have misunderstood the model).

14) check sense of line 254

15) with respect to role and modelling of ion flux, authors might note recent paper from the Blatt group in Nature Plants (2021).

---

## [Reviewer Report]

Dear Editor,

We are submitting this revised original research article, entitled “Turgor pressure changes in stomatal guard cells arise from interactions between water influx and mechanical responses of their cell walls.” We reviewed and responded to all the comments in the revised manuscript. As noted in the response, we believe that the revision addresses insightful and detailed comments of reviewers resulting in a strengthened and balanced manuscript. We look forward to your reply.

Sincerely,

Dr. Hojae Yi and Dr. Charles T. Anderson

The Pennsylvania State University

---

## [Reviewer Report]

*Comments to Author*: The authors have addressed all comments raised in the first round of review.

---

## [Reviewer Report]

*Comments to Author*: There are a few typos that need correcting in the manuscript, but otherwise the authors have done a good job of addressing the points raised in the review process.

---

## [Reviewer Report]

*Comments to Author*: 

Dear Authors, 

Both reviewers and I think you have done a very thorough job at addressing the comments and points that were previously raised. Congratulations on the acceptance of this excellent and interesting manuscript. I am delighted you chose quantitative plant biology for your work. With thanks and best wishes

Richard